# Design of effective value calculation model for dynamic dataflow of infrared gas online monitoring

**Dong Xiao**[1,2]*, **Lu Huang**[3], **Mohamed Keita**[1,4], **Hailun He**[5], **Dayong Chen**[1], **Jin Li**[6]*

**1** CUMT-UCASAL Joint Research Center for Biomining and Soil Ecological Restoration, State Key Laboratory of Coal Resources and Safe Mining, China University of Mining and Technology, Xuzhou, Jiangsu, China, **2** State Key Laboratory of Coal Resources and Safe Mining, China University of Mining and Technology (Beijing), Beijing, China, **3** School of Management, Xiamen University, Xiamen, Fujian, China, **4** School of Mines, China University of Mining and Technology, Xuzhou, Jiangsu, China, **5** School of Life Science, Central South University, Changsha, Hunan, China, **6** Department of General Surgery, Xuzhou First People's Hospital, Affiliated Hospital of China University of Mining and Technology, Xuzhou, Jiangsu, China

* xd@cumt.edu.cn (DX); lijin_tjdoctor@163.com (JL)

**Data Availability Statement:** All relevant data are within the manuscript and its Supporting Information files.

**Funding:** This work was supported by The Open Research Project of State Key Laboratory of Coal

## Abstract

The development of "CC30A $CH_4$-$CO_2$ combined analyzer" with infrared gas sensor as the core detection device can be widely used in online gas component analysis. In data analysis, the maximum value and arithmetic mean of the sensor data for each test period are not effective value. The characteristics of the dynamic data are: (1) Each DAW completes one test for one parameter, there is a unique effective value; (2) In test state, the fluctuation of the sensor value gradually decreases when approaching to the end of the test. An effective value calculation model was designed using the method of dimensionality reduction of dynamic data. The model was based on the distribution characteristics of the process data, and consists of 4 key steps: (1) Identify the Data Analysis Window (DAW) and build DAW dataset; (2) Calculate the value of optimal DAW dataset segmentation and build DAW subdataset; (3) Calculate the arithmetic mean ($M_c$) and count the amount of data in each subdataset ($F_c$), and build the optimal segmentation statistical set; (4) Effective value calculation and error evaluation. Calculation result with 50 sets of monitor data conformed that the EVC model for dynamic data of gas online monitoring meets the requirements of experimental accuracy requirements and test error. This method can be independently calculated without relying on the feedback information of the monitoring device, and it has positive significance for using the algorithm to reduce the hardware design complexity.

## Introduction

$CH_4$ and $CO_2$ are the key products in the anaerobic fermentation of coal or biomass, and the trend analysis of gas composition changes is the key factor for the control of anaerobic digestion. Gas composition online analysis technology has gradually diversified with the improvement of the accuracy of gas sensors [1]. Especially using infrared $CH_4$-$CO_2$ gas sensor instead

Resources and Safe Mining (grant number SKLCRSM17KFA08: DX), The Independent Research Project of State Key Laboratory of Coal Resources and Safe Mining, CUMT (grant number SKLCRSM19X0012: DX), The Key Research and Development Project of Xuzhou Science and Technology (grant number KC20137: JL). D.X. and J.L. wrote and review this paper.

**Competing interests:** The authors have declared that no competing interests exist.

**Abbreviations:** GC, Gas Chromatography; EVC, Effective Value Calculation Model; TCD, Thermal Conductivity Detector; DND, Data Numerical Distribution; DFDS, Data Frequency Distribution of Subdataset; $S_i$, segmentation value; DAW, Data Analysis Window; $C_t$, Threshold Value; $SN_b$, the amount of data in number b subdataset; OSS, Optimal Segmentation Statistical Set; VD, value dispersion.

of gas chromatography (GC) to achieve low-cost online gas components detection and analysis [2, 3]. At the same time, the optimization of data analysis methods (such as profile monitoring technology [4, 5] and ensemble model [6]) and the improvement of computing capabilities of single-chip microcomputer, has further improved the reliability and accuracy of online monitoring [7]. When the sensor is working, it analyzes the target gas concentration in its gas chamber in real time, and continuously send the value to microcomputer [8], forming a data flow which is named dynamic data flow. However, in order to avoid the mutual interference of different gas samples and ensure the test accuracy, the analyzer will flush gas chamber with $N_2$ or air after complete each test [9]. Meanwhile, the response time of sensor further enhanced the delay effect of test result [10]. For sensors with quick response time, such as noncontact thermopile [11], pressure difference sensor [12], hall sensor [13], etc., the raw sensor data is correlated with the measured parameter and identified effective data, when the data collection period is longer than the sensor response period. And for the sensor with slow response time (such as infrared $CH_4$ sensors [14], the response time longer than 20 S), the effective data will be mixed with process data when the data collection frequency is shorter than the sensor response period is. Infrared $CH_4$-$CO_2$ sensor is a typical case. These two factors cause the gas composition to be in dynamic changes all the time in sensor gas chamber, and effective data and process data mixes in dynamic data flow. More than 80% of the sensor data is process data, which is invalid data, when data collection frequency is set to 1 data/S.

In the gas composition online analysis technology, the tube bundle system is widely used in coal mines [15, 16], anaerobic digestion systems [17, 18], and fermentation industries [19]. This system uses one set/group of gas sensors to perform gas concentration tests from multiple monitoring points in patrol read mode [16]. And it suitable to test the atmosphere where the gas composition is relatively stable and the gas concentration changes slowly. $CH_4$ and $CO_2$ yield monitoring of coal biogasification experiment are a typical case of tube bundle system utilization [20]. Affected by the characteristics of the biodegradation of organic compounds in coal, the experiment cycle is usually more than 200 days, or even one year [21, 22]. When setting the test frequency for each experiment sample to 8 hours, one system can serve more than 60 samples. Gas analysis measures 80 experiment samples in a patrol model, and outputs one set of data every second (take the IR-EK2 infrared $CH_4$-$CO_2$ sensor evaluation kit as an example). The total amount of data can reach 1.66 billion for one standard coal biogasification experiment, if all the data would be recorded. However, only 1.4% of the data is effective data. The storage of a large amount of process data is not only waste storage resources, but also waste computing resources for data analysis. Meanwhile, limited by the computing power of the ARM7 processor [23], it is necessary to design an algorithm that occupies less computing resources to ensure that the processor has enough computing power to process concurrent instructions. Therefore, the dynamic dataflow analysis and effective value calculation would be important for simplifying data processing and improving the utilization of computing resources.

## Methods

### Gas chromatographic analysis of gas composition

1.00mL of gas samples were collected from every experiment sample in each test cycle. And the $CH_4$ and $CO_2$ contents were analyzed by gas chromatography (GC) (7890A, Agilent, America). The gas composition tested by GC was defined as GC value.

The $N_2$ (carrier gas) flow rate was set at 1.0 mL/min. The injection port was maintained at 150°C, the oven temperature was 25°C, and the thermal conductivity detector (TCD) was operated at 200°C [24]. The retention time for methane was 3.76 minutes, and for carbon

dioxide was 5.0 minutes. Calibration standards consisting of 40% $CH_4$, 20% $CO_2$, 10% $H_2$ and 30% $N_2$ were injected to generate the calibration plot.

### Gas composition online monitoring

The gas composition analysis flow data was monitored using CC30A $CH_4$-$CO_2$ combined analyzer (Jundong, China). The gas chamber was 5.0 mL. The analysis period was set for 8 hours, one injection volume was set to 90 mL, the flow rate was set to 120 mL/min, and the dehydrator temperature was set to -25°C. The gas sample time was set to 45 S, and the test time was set to 90 S. After one test was completed, the sensor was flushed with $N_2$ for 60 S and with dried air for 165 S. $N_2$ and air flow rate was set to 100 mL/min. The CC30A sensitivities of $CH_4$ and $CO_2$ were 2000 ppm, and the resolution was 500 ppm for both gases.

## Effective value analysis model design for gas online monitoring

### Dynamic data characteristics of gas online test

In the coal biogasification experiment, "CC30A $CH_4$-$CO_2$ combined analyzer (abbreviated as CC30A)" was used to test the gas composition of each sample every 8 hours. When CC30A was in preparation state before the gas test, the gas chamber was filled with air, and the sensor data of $CH_4$ and $CO_2$ below 0.50%vol. This data was identified as background value.

The sensor data in test state, 90 mL gas sample was slowly injected into the gas chamber, and $CH_4$ and $CO_2$ concentration in the gas chamber was gradually increased. The gas sample injection was lasted for 45 S, the gas composition in the gas chamber gradually consistent with the sample (Fig 1. Stage), and the slope of curve were gradually decreased. When gas sample injection process was over, the gas was sealed in gas chamber for 90 S.

When the sensor data was tending to be stable and the test result was tending to be consistent with the GC value (Fig 1. Stage ). After the test state was completed, the gas chamber was (why would be) flushed with $N_2$ for 60 seconds (Fig 1. Stage ). At the beginning of stage , affected by the check valve, which set at the air outlet of gas chamber, in the initial stage of $N_2$ flushing, the short-term pressure increase in the chamber caused the sensor value to increase first and then decrease (Fig 1, stage ). Stage  to stage  constitute a data analysis window (DAW). According to the CC30A $CH_4$-$CO_2$ combined analyzer design, in the later stage of stage , the sensor data tended to be stable and fluctuated around the effective value. Therefore, according to the data distribution characteristics in a monitoring window, the effective value is equal to the sensor value with the highest frequency. Because CC30A completes the gas concentration of $CH_4$ and $CO_2$ simultaneously, two gas test values have the same data fluctuation. To simplify the calculation of effective value, monitoring window should be identified firstly in dynamic data.

The data in Fig 1 as an example, the DAW was established based on $CH_4$ test results, and the threshold value ($C_t$) was defined as 0.50% (background value). When the dynamic data was larger than the $C_t$, and data fluctuation conformed to stages - , this data set was defined as one Data Analysis Window (DAW). The effective value calculation model was designed based on the window.

According to the $CH_4$ sensor data distribution in Fig 1, compare the character difference of data in stage /  and stage , it was: (1) The absolute value of the difference between adjacent data was larger in stage and ; (2) If divided the $CH_4$ test value from $C_t$ to the maximum into several equal parts, the amount of data in the interval of effective value was the largest. Therefore, two concepts were defined in the calculation model design: (1) Data Numerical Distribution (DND): the absolute value of the difference between adjacent data in one monitoring window data set; (2) Data Frequency Distribution of Subdataset (DFDS): the sensor value from $C_t$ to

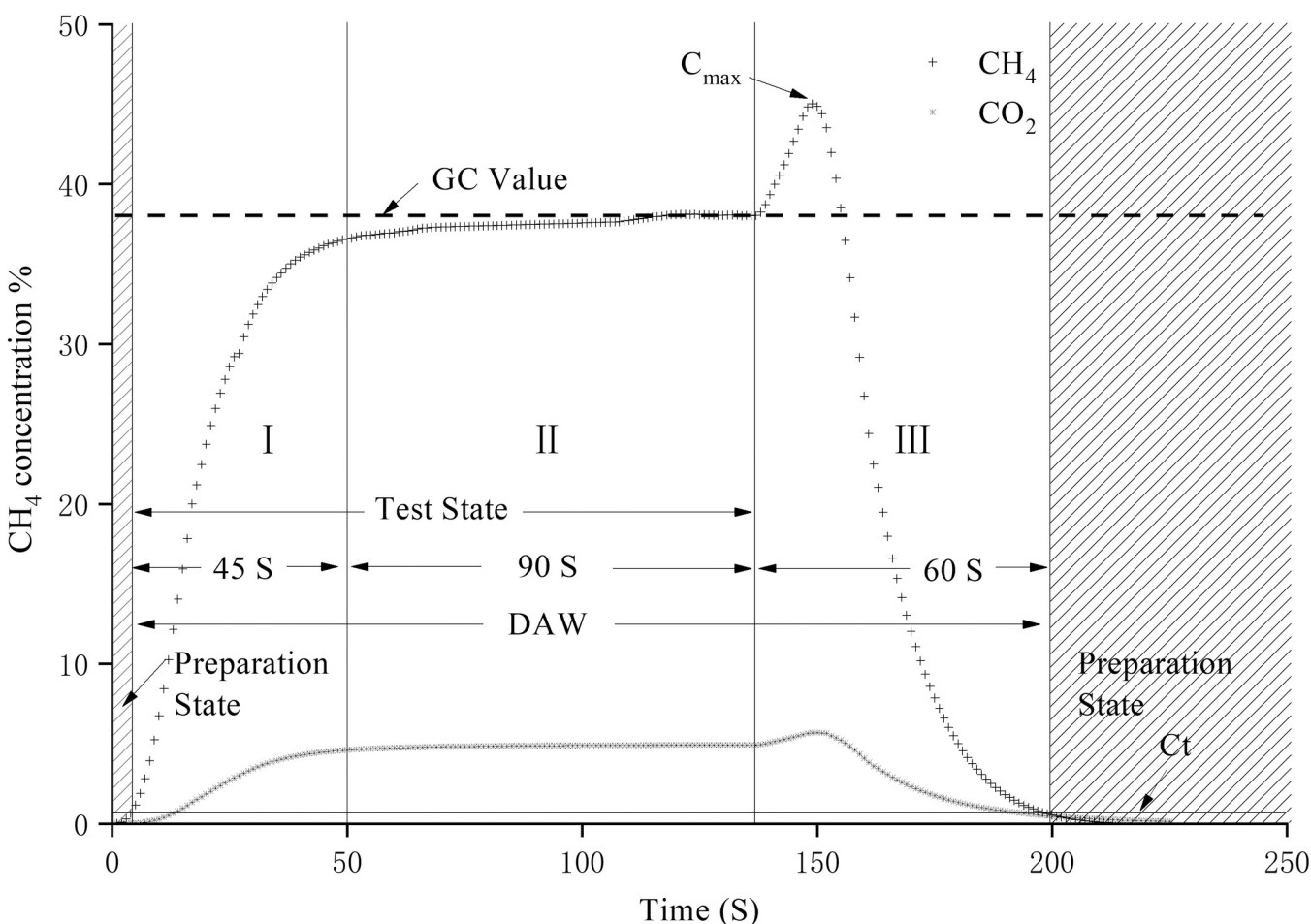

**Fig 1. Gas concentration curve in one monitoring window.** The test process was divided into three stages based on the test method and gas concentration change factors. Ct is the threshold value.

the maximum in one DAW was divided into several subdataset. DFDS is the amount of data in every subdataset if the data was simplified according to certain hexadecimal, the data in the DAW would be divided into groups. There should be a group of dataset with the highest statistical value, and the effective value should be in this group. By this data analysis method, the effective value could be identified with the data value domain segmentation (DVDS) and statistical frequency (SF) analysis.

### Effective value calculation model design of dynamic dataflow

Effective value calculation was based on a data set in one DAW. The characteristics of every dataset were: (1) each DAW completes one test for one parameter, there is a unique effective value; (2) in test state, the fluctuation of the sensor value gradually decreases when approaching the end of the test. Therefore, the frequency of each sensor data in stage and  was less than 2; meanwhile, the frequency of some sensor data in stage was higher than 2. The effective value calculation model was designed as follow steps:

(1) Identify the DAW in the dynamic dataflow.

One gas concentration test cycle was 360 S. The CC30A sensitivities of $CH_4$ and $CO_2$ were 2000 ppm, and the resolutions were 500 ppm for both gases. Threshold value ($C_t$) was been

used to identify the test data and value was set to 0.50%. The first sensor data greater than $C_t$ was defined as the DAW initial data. The time of each DAW was marked by the time of the DAW initial data plus 135 S. In order to prevent misjudgment caused by abnormal data fluctuations, it was required that 100 sensor data before the DAW initial data should less than $C_t$ and the data volume of DAW data set no less than 150.

(2) Optimal DAW dataset segmentation.

In the second step, a DAW dataset was divided from $C_t$ to the maximum data ($C_{max}$) into n equal parts (n DAW subdatasets, and n is calculated by Eq 1) (Fig 2). The value of segmentation was integer multiple of CC30A resolution ($S_i$), and the multiple was limited 1–20 (example: when the multiple value was 2, the segmentation value was recorded as $S_2$). The smaller value of i, the larger value of n, and the greater number of subdatasets. The value of i start from 1, and count the amount of data in each subdataset ($SN_b$) under the $S_i$ until the $SN_b$ has a unique maximum. Through a large amount of data analysis, the $SN_{b-max}$ should be greater than 6 in order to reduce the effective value error.

Fig 3 as an example, when the segmentation value of i was 1, the $SN_{b-max}$ of two subdatasets was 3 (Fig 3A-1); when the value of i was 2, there was only one $SN_{b-max}$ = 4 (Fig 3B-1). Until the value of i was 10, there was only one $SN_{b-max}$ = 7 (Fig 3C-1) Therefore, the optimal value of DAW dataset segmentation was $S_{10}$.

(3) The optimal segmentation statistical set building.

Based on DAW dataset segmentation, statistical calculation of the average value of each subdataset and DFDS values.

Step 1: Build DAW dataset. After the DAW has been established, removed the time stamp and reordered the sensor data in the window ($V_1 \ldots \ldots V_m$) (Step 1 in Fig 2).

Step 2: Calculate the value of optimal DAW dataset segmentation. And divide the DAW dataset into n equal parts, build DAW subdataset ($SN_b$, b = 1 to n).

$$n = \frac{C_{max}}{S_i} \tag{1}$$

where: n: the number of DAW subdataset;

$C_{max}$: the largest value in DAW;

Si: segmentation value

Step 3: calculate the arithmetic mean ($M_c$) and count the amount of data in each subdataset ($F_c$). If there is no data in a subdataset, it is judged as an invalid subset and recorded as empty. The empty subset would not record into Optimal Segmentation Statistical Set (OSS).

$$M_c = \frac{\sum_{i=b_{min}}^{b_{max}} V_i}{(b_{max} - b_{min}) + 1} \tag{2}$$

Where: $M_c$: the c data in Optimal Segmentation Statistical Set;

$V_i$: a data in a DAW dataset;

$b_{max}$ is the max $V_i$ number in a DAW subdataset;

$b_{min}$ is the min $V_i$ number in a DAW subdataset;

Step 4: Build the optimal segmentation statistical set. There are two parameters in each OSS subset: the arithmetic mean of $SN_b$, number of data in $SN_b$.

(4) Effective value calculation.

Effective value is one data in optimal segmentation statistical set. It needs meet two conditions at the same time: it has the unique maximum SN; the effective value is no the largest OSS data. Therefore, the effective value can be found by the query method. A review calculation was introduced in this model to ensure the reliability of effective value. The principle was

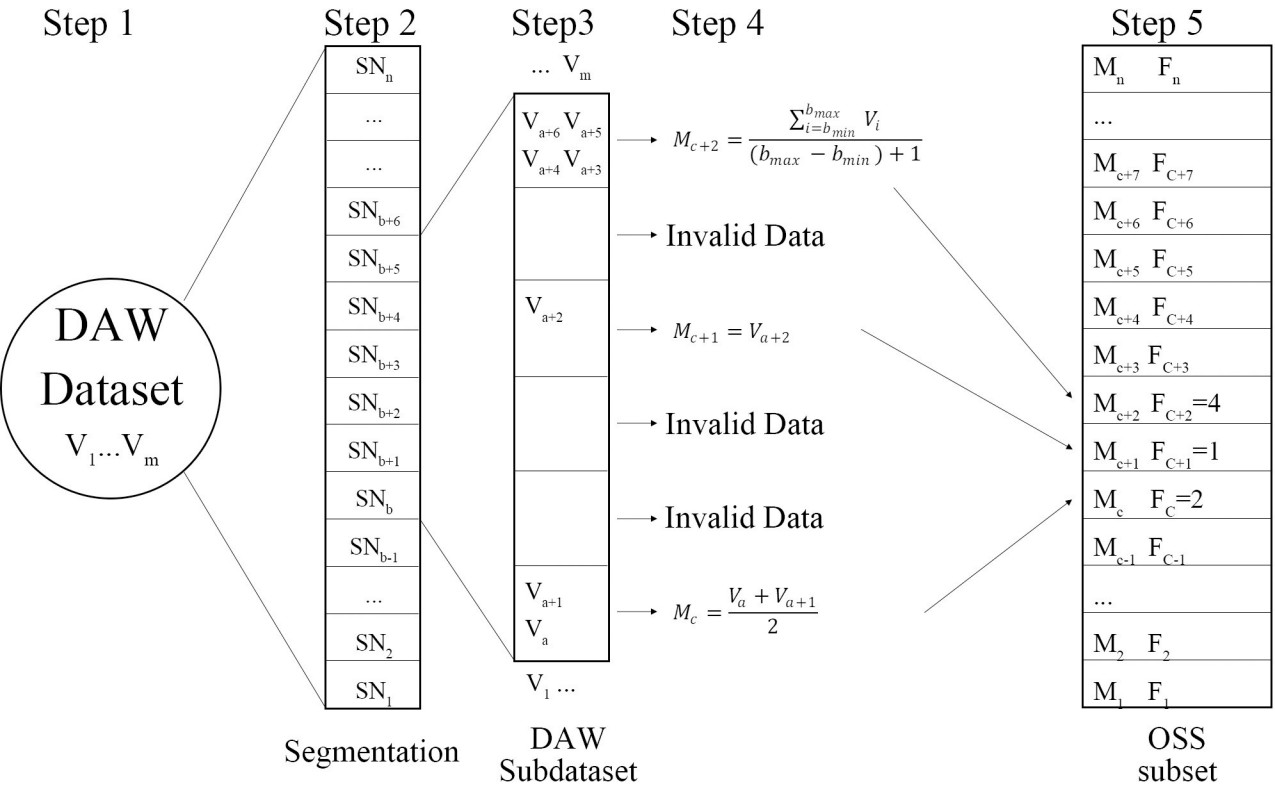

**Fig 2. Monitoring window data set segmentation and effective segmentation statistical set calculation method.**

that the curve slope where the effective value was located tends to 0. The value dispersion (VD) was introduced, and it could analyze the proportional relationship between the adjacent data of effective value and optimal DAW dataset segmentation (Eq 3). Because the fluctuation of sensor data at the end of test was the smallest, the VD of effective value should close to 1. Taking the data of Fig 1 as an example, the VD was large in the stage and  and small in the Stage , especially for the data close to the effective value (Fig 3A-2, 3B-2 and 3C-2).

$$\text{VD}_b = \frac{(OSS_{b+1} - OSS_{b-1})}{2S_0} \quad (b - 1 \geq 1) \tag{3}$$

Where: $\text{VD}_b$: the value dispersion of the b data;
$OSS_b$: the b data in optimal segmentation statistical set;
$S_0$: the optimal DAW dataset segmentation.

## Validation of the effective value calculation model

The effective value calculation (EVC) model was verified based on 50 sets of $CH_4$ and $CO_2$ data which obtained in the coal biogasification experiment. The calculation process was performed according to the data acquisition time. And the error of each parameter was calculated based on GC value. The allowable error was set to≤2.5% (Fig 4).

Calculation result conformed that the EVC model for dynamic data of gas online monitoring less than allowable error. The relative error statistics data identified that the relative error for more than 65% data was less than 1% for both $CH_4$ and $CO_2$ concentration data.

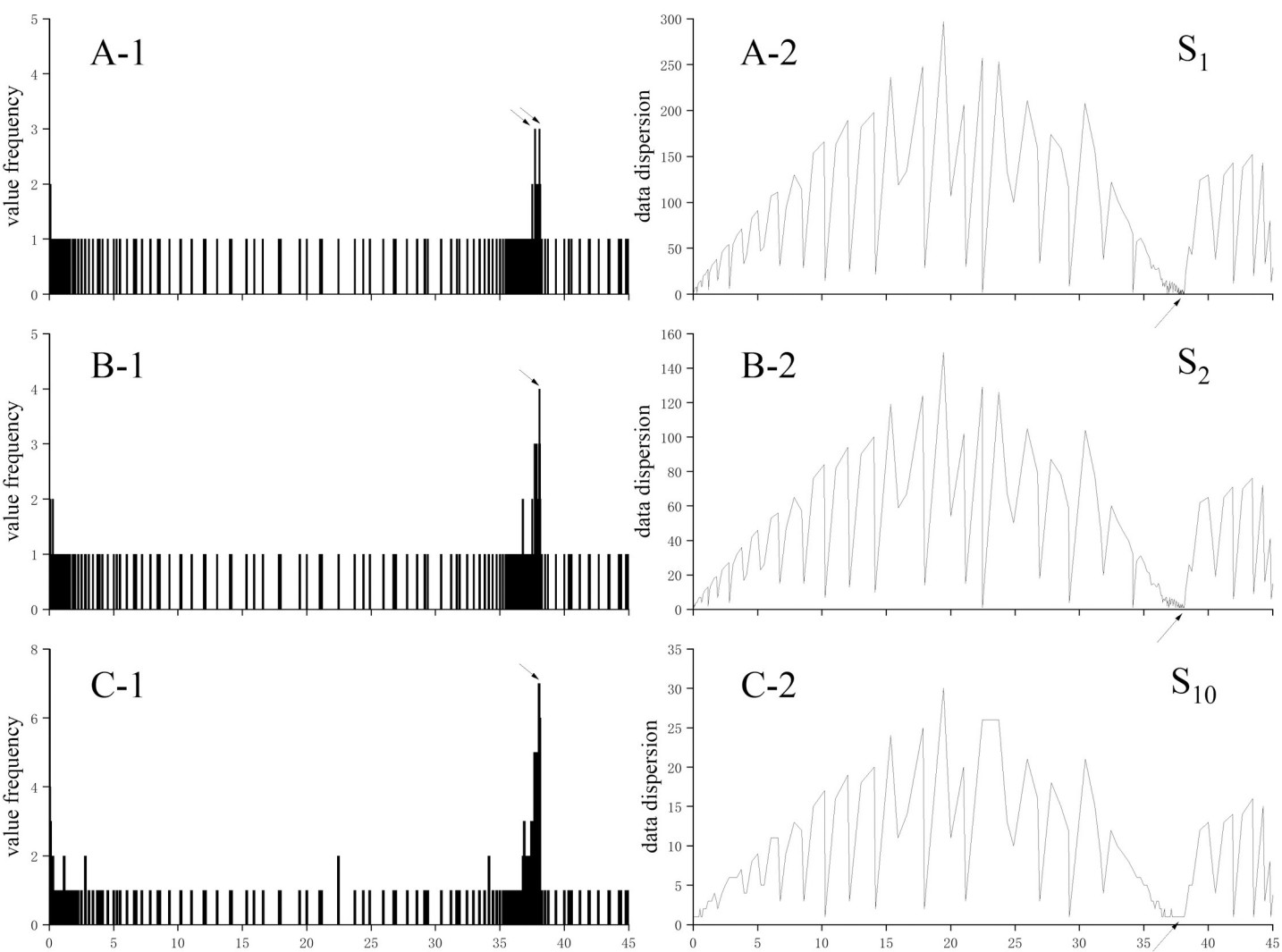

**Fig 3. Value frequency and data dispersion changes with difference value domain segmentation.** Fig A shows the variation of the value frequency of different segmentation values with a histogram. Fig B shows the data dispersion for different segmentation values. The smaller the dispersion, the better the relative continuity of the values in the data set.

## Results and discussion

This model was designed for the key algorithm in CC30A $CH_4$-$CO_2$ combined analyzer system development. In the CC30A system, infrared $CH_4$-$CO_2$ sensors were used as the core analysis unit. The system needs to be physical separated between calculation unit and analysis unit, including power ground and signal transmission. To solve this problem, only one optical coupler was used in system design to establish one signal isolation path for the two units. This simplified design improved the anti-interference ability of the system, but created a problem which was how to make the computing unit fast and complete the result analysis with low calculate resource utilization. This algorithm design was based on the analysis steps and sensor characteristics of CC30A $CH_4$-$CO_2$ combined analyzer, and clarifies the reason and law of the fluctuation of the infrared gas component sensor data. It was to allow the computing unit to automatically lock the data analysis window and complete the effective value calculation according to data changes in dynamic data flow.

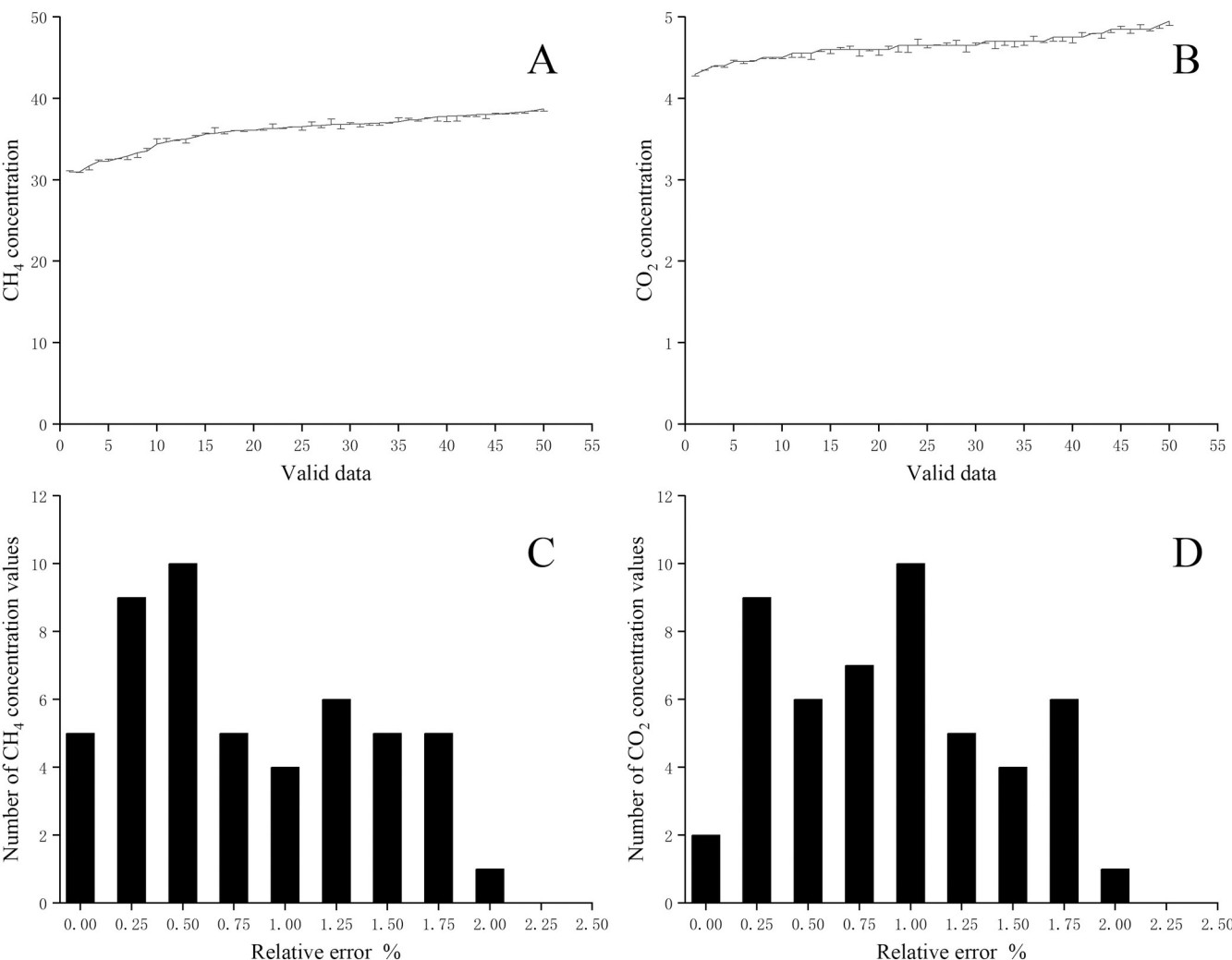

**Fig 4. The absolute error and relative error statistics of effective value analysis model calculation results.** Fig A and B are the effective value analysis model calculation result with absolute error for 50 sets of $CH_4$ and $CO_2$ data. Fig C and D show the statistical results of the data in graphs A and B respectively with 0.5% relative error interval.

The model uses the data dimensionality reduction method to extract the data distribution characteristics. The dimensionality reduction calculation instead of curve slope change analysis or curve fitting, and completes the calculation of the effective value by the conditional judgment and mean calculation within 8 data. This algorithm calculation is mainly divided into three steps (Fig 5). The calculation of the DAW subdataset and the OSS subset completed the dimensionality reduction calculation of the original data. It is the core algorithm of the EVC model. The model was verified based on 50 sets of $CH_4$ and $CO_2$ data. Error analysis confirmed that the EVC model for dynamic data of gas online monitoring meets the requirements of experimental accuracy requirements and test error.

## Conclusion

Real-time and accurate acquisition the concentration of $CH_4$ and $CO_2$ in anaerobic fermentation is key indicators for monitoring the fermentation system. The design goal of the CC30A $CH_4$-$CO_2$ combined analyzer is to realize low-cost, automated, real-time online analysis of key

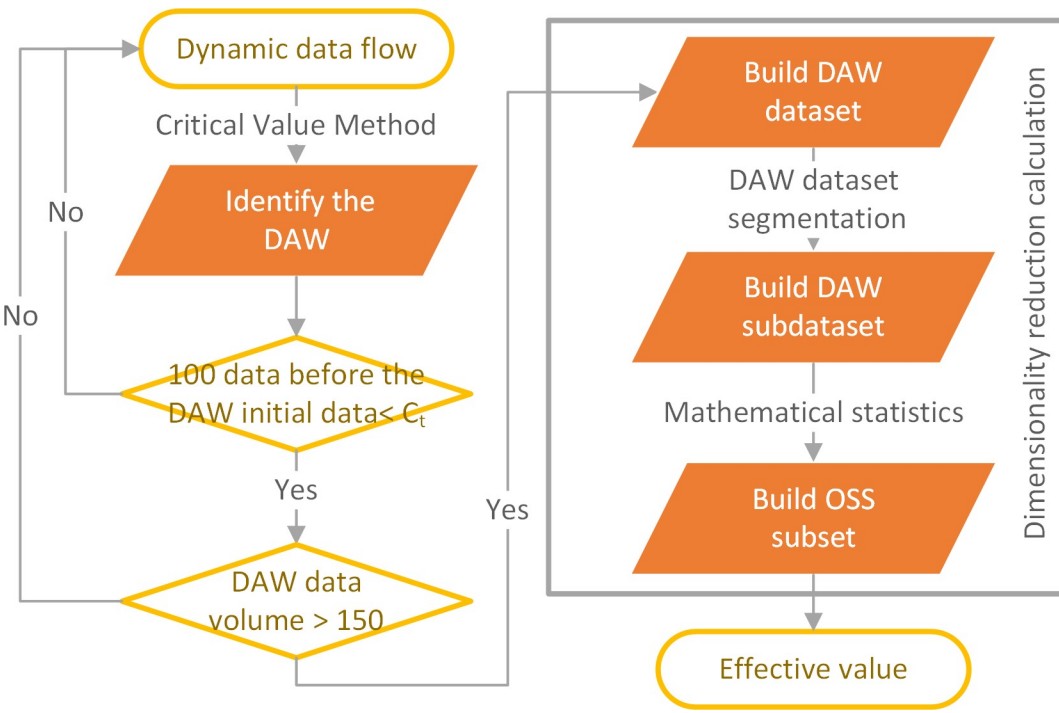

**Fig 5. Effective Value Calculation (EVC) model calculation flow chart.**

gas components. Limited by the computing power of the ARM7 processor, the effective value calculation (EVC) model has been designed. The advantage of this algorithm is that only a few simple judgments and statistics are needed to replace complex algorithms to extract the target data from the data flow. According to the principle of model operation, the model can be well utilized to dynamic data flow operations with the following characteristics. (1) the effective value is related to the data distribution characteristics, and is not the maximum or average value in dataset; (2) the calculation is independent and complete according to the fluctuation of the data, and does not rely on any peripheral devices signals. The design of the EVC model enables the calculations independently, and it has positive significance for using the algorithm to reduce the hardware design complexity.

## Supporting information

**S1 Dataset. Original origin data of Fig 1.**
(OPJU)

**S2 Dataset. Original origin data of Fig 3.**
(OPJU)

**S3 Dataset. Original origin data of Fig 4.**
(OPJU)

**S4 Dataset. Raw data in paper analysis.**
(RAR)

**S1 Definition.**
(DOCX)

## Acknowledgments

The authors' acknowledge the contributions of the following companies for allowing access to coal samples and other information used in this paper: Sihe mining, J&D Technology Company.

## Author Contributions

**Data curation:** Dong Xiao, Lu Huang, Hailun He, Dayong Chen.

**Funding acquisition:** Dong Xiao, Jin Li.

**Methodology:** Lu Huang.

**Writing – original draft:** Dong Xiao, Lu Huang.

**Writing – review & editing:** Mohamed Keita, Hailun He, Dayong Chen, Jin Li.

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
