## [Decision Letter · Decision Letter 0]

28 Aug 2021

PONE-D-20-38017

Design of Effective Value Calculation Model for Dynamic Dataflow of Infrared Gas Online Monitoring

PLOS ONE

Dear Dr. Xiao,

Thank you for submitting your manuscript to PLOS ONE. After careful consideration, we feel that it has merit but does not fully meet PLOS ONE’s publication criteria as it currently stands. Therefore, we invite you to submit a revised version of the manuscript that addresses the points raised during the review process.

Please pay particular attention to carefully addressing the points raised by Reviewer 1. Regarding the references suggested by Reviewer 2, please only include these if you feel they are relevant to the context of your study.

We look forward to receiving your revised manuscript.

Kind regards,

Jamie Males

Staff Editor

PLOS ONE

Journal Requirements:

Reviewers' comments:

Reviewer's Responses to Questions

**Comments to the Author**

1. Is the manuscript technically sound, and do the data support the conclusions?

Reviewer #1: Yes

Reviewer #2: Yes

2. Has the statistical analysis been performed appropriately and rigorously? 

Reviewer #1: Yes

Reviewer #2: Yes

3. Have the authors made all data underlying the findings in their manuscript fully available?

Reviewer #1: Yes

Reviewer #2: Yes

4. Is the manuscript presented in an intelligible fashion and written in standard English?

Reviewer #1: Yes

Reviewer #2: No

5. Review Comments to the Author

Reviewer #1: The paper is well written and interesting. I recommend accepting it after a minor revision.

1. Please note the standardization of writing. For example, there should be no colon after "results and discussion".

2. Literature review is insufficient. I suggest that the author read the following papers and add them to the literature review:

(1) Lu, H., Iseley, T., Matthews, J., Liao, W., & Azimi, M. (2021). An ensemble model based on relevance vector machine and multi-objective salp swarm algorithm for predicting burst pressure of corroded pipelines. Journal of Petroleum Science and Engineering, 203, 108585.

(2) Xu, Z. D., Yang, Y., & Miao, A. N. (2021). Dynamic Analysis and Parameter Optimization of Pipelines with Multidimensional Vibration Isolation and Mitigation Device. Journal of Pipeline Systems Engineering and Practice, 12(1), 04020058.

(3) Lu, H., Iseley, T., Behbahani, S., & Fu, L. (2020). Leakage detection techniques for oil and gas pipelines: State-of-the-art. Tunnelling and Underground Space Technology, 98, 103249.

Reviewer #2: This study attempts to explain design of effective value calculation model for dynamic data-flow of infrared gas online monitoring. The study develops an effective value calculation model using the method of dimensionality reduction of dynamic data. The model was based on the distribution characteristics of the process data, and consisted of four steps. This research aids in improving knowledge on gas monitoring and should be considered for publication after addressing the following suggestions / comments:

1. The introduction section of the study should include the background of the study and the concept of dynamic data flow analysis for clarity on the subject. This is necessary to offer an understanding of the topic and scope of the study.

2. The results and discussion section should be enhanced further. In fact, it primarily offers a summary and repetition of information already appearing in both introduction and methods section. Therefore, this section needs to be revised to offer a discussion of the study results.

3. The study lacks conclusion section. This is an integral part of any research and need to be added.

4. Additionally, the following style and grammatical errors should be addressed:

i. Check the in-text references and make sure they are according to the author guideline. Avoid any mixing two different referencing styles. For example, in line 71 there is a superscript 9. Is this for the foot notes or a typo?

ii. Ensure that you use correct symbols all through the paper. Example, check line 103

iii. Follow the provided author guidelines while numbering and formatting your equations

iv. Kindly proof read your document before submission to ensure that all sentences are well structured.

6. PLOS authors have the option to publish the peer review history of their article (what does this mean?). If published, this will include your full peer review and any attached files.

Reviewer #1: No

Reviewer #2: **Yes: **Joseph Githiria

---

## [Author Response · Author response to Decision Letter 0]

13 Sep 2021

REVIEWER #1: 

The paper is well written and interesting. I recommend accepting it after a minor revision.

Q1. Please note the standardization of writing. For example, there should be no colon after "results and discussion".

Answer：wither another reviewer's suggestion, a “conclusion” chapter was added after "results and discussion".

The chapter 6“6. Declarations” has been modified as “agreements” according to the PLOS ONE specification.

Q2. Literature review is insufficient. I suggest that the author read the following papers and add them to the literature review:

(1) Lu, H., Iseley, T., Matthews, J., Liao, W., & Azimi, M. (2021). An ensemble model based on relevance vector machine and multi-objective salp swarm algorithm for predicting burst pressure of corroded pipelines. Journal of Petroleum Science and Engineering, 203, 108585.

(2) Xu, Z. D., Yang, Y., & Miao, A. N. (2021). Dynamic Analysis and Parameter Optimization of Pipelines with Multidimensional Vibration Isolation and Mitigation Device. Journal of Pipeline Systems Engineering and Practice, 12(1), 04020058.

(3) Lu, H., Iseley, T., Behbahani, S., & Fu, L. (2020). Leakage detection techniques for oil and gas pipelines: State-of-the-art. Tunnelling and Underground Space Technology, 98, 103249.

Answer: These three articles are very interesting and meaningful. They have been added in references.

REVIEWER #2: 

This study attempts to explain design of effective value calculation model for dynamic data-flow of infrared gas online monitoring. The study develops an effective value calculation model using the method of dimensionality reduction of dynamic data. The model was based on the distribution characteristics of the process data, and consisted of four steps. This research aids in improving knowledge on gas monitoring and should be considered for publication after addressing the following suggestions / comments:

1. The introduction section of the study should include the background of the study and the concept of dynamic data flow analysis for clarity on the subject. This is necessary to offer an understanding of the topic and scope of the study.

Answer: the introduction section has been rewritten, and add background of this study (as follows). The definition of dynamic data flow has been added in “Definition” and “Introduction” section.

CH4 and CO2 are the key products in the anaerobic fermentation of coal or biomass, and the trend analysis of gas componsition changes is the key factor for the control of anaerobic digestion. Gas composition online analysis technology has gradually diversified with the improvement of the accuracy of gas sensors[1]. For example, using infrared CH4-CO2 gas sensor instead of gas chromatography (GC) to achieve low-cost online gas components detection and analysis[2,3]. At the same time, the optimization of data analysis methods (such as profile monitoring technology[4,5] and ensemble model[6]) and the improvement of computing capabilities of single-chip microcomputer, has further improved the reliability and accuracy of online monitoring. When the sensor is working, it analyzes the target gas concentration in its gas chamber in real time, and continuously send the value to microcomputer, forming a data flow which is named dynamic data flow. However, taking the CC30A CH4-CO2 combined analyzer (Jundong, China) as an example, in order to avoid the mutual interference of the gas samples from each channel and ensure the test accuracy, the analyzer will flush gas chamber with N2 after complete each test. Meanwhile, the response time of sensor further enhanced the delay effect of test result. For sensors with quick response time (such as noncontact thermopile[7], pressure difference sensor [8], hall sensor[9], etc.), the raw sensor data is correlated with the measured parameter and identified effective data, when the data collection period is longer than the sensor response period. And for the sensor with slow response time (such as infrared CH4 sensors[10], the response time longer than 20 S), the effective data will be mixed with process data when the data collection frequency is much shorter than the sensor response period is. Infrared CH4-CO2 sensor is a typical case. These two factors cause the gas composition to be in dynamic changes all the time in sensor gas chamber, and effective data and process data mixes in dynamic data flow. More than 80 % of the sensor data is process data, which is invalid data, when data collection frequency is set to 1 data/S.

Q2. The results and discussion section should be enhanced further. In fact, it primarily offers a summary and repetition of information already appearing in both introduction and methods section. Therefore, this section needs to be revised to offer a discussion of the study results.

Answer：the result and discussion have been rewritten, and add a new figure (Figure 5) has been added.

This model was designed for the key algorithm in CC30A CH4-CO2 combined analyzer system development. In the CC30A system, infrared CH4-CO2 sensors were used as the core analysis unit. The system needs to be physical seperated between calculation unit and analysis unit, including power ground and signal transmission. To solve this problem, only one optical coupler was used in system design to establish one signal isolation path for the two units. This simplified design improved the anti-interference ability of the system, but created a problem which was how to make the computing unit fast and complete the result analysis with low calculate resource utilization. This algorithm design was based on the analysis steps and sensor characteristics of CC30A CH4-CO2 combined analyzer, and clarifies the reason and law of the fluctuation of the infrared gas component sensor data. It was to allow the computing unit to automatically lock the data analysis window and complete the effective value calculation according to data changes in dynamic data flow. 

The model uses the data dimensionality reduction method to extract the data distribution characteristics. The dimensionality reduction calculation instead of curve slope change analysis or curve fitting, and completes the calculation of the effective value by the conditional judgment and mean calculation within 8 data. This algorithm calculation is mainly divided into three steps (Figure 5) . The calculation of the DAW subdataset and the OSS subset completed the dimensionality reduction calculation of the original data. It is the core algorithm of the EVC model. The model was verified based on 50 sets of CH4 and CO2 data. Error analysis confirmed that the EVC model for dynamic data of gas online monitoring meets the requirements of experimental accuracy requirements and test error. 

Figure 5 here

Figure 5 effective value calculation (EVC) model calculation flow chart.

Q3. The study lacks conclusion section. This is an integral part of any research and need to be added.

Answer：This suggestion is very important, the “Conclusion” section has been added.

6. Conclusion

Real-time and accurate acquisition the concentration of CH4 and CO2 in anaerobic fermentation is key indicators for monitoring the fermentation system. The design goal of the CC30A CH4-CO2 combined analyzer is to realize low-cost, automated, real-time online analysis of key gas components. Limited by the computing power of the ARM7 processor, the effective value calculation (EVC) model has been designed. The advantage of this algorithm is that only a few simple judgments and statistics are needed to replace complex algorithms to extract the target data from the data flow. According to the principle of model operation, the model can be well utilized to dynamic data flow operations with the following characteristics. (1) the effective value is related to the data distribution characteristics, and is not the maximum or average value in dataset; (2) the calculation is independent and complete according to the fluctuation of the data, and does not rely on any peripheral devices signals. The design of the EVC model enables the calculations independently, and it has positive significance for using the algorithm to reduce the hardware design complexity.

Q4. Additionally, the following style and grammatical errors should be addressed:

i. Check the in-text references and make sure they are according to the author guideline. Avoid any mixing two different referencing styles. For example, in line 71 there is a superscript 9. Is this for the foot notes or a typo?

Answer: mistakes of referencing styles have been revised.

ii. Ensure that you use correct symbols all through the paper. Example, check line 103

Answer: symbols mistakes have been revised in paper.

iii. Follow the provided author guidelines while numbering and formatting your equations

Answer: all the equations and numbering have been re-edit.

iv. Kindly proof read your document before submission to ensure that all sentences are well structured.

Answer：The grammar and presentation of the thesis have been modified by native English students.

---

## [Editor Report · Decision Letter 1]

14 Oct 2021

Design of Effective Value Calculation Model for Dynamic Dataflow of Infrared Gas Online Monitoring

PONE-D-20-38017R1

Dear Dr. Xiao,

We’re pleased to inform you that your manuscript has been judged scientifically suitable for publication and will be formally accepted for publication once it meets all outstanding technical requirements.

Kind regards,

Pasquale Avino, Ph.D.

Academic Editor

PLOS ONE

---

## [Editor Report · Acceptance letter]

19 Oct 2021

PONE-D-20-38017R1 

Design of Effective Value Calculation Model for Dynamic Dataflow of Infrared Gas Online Monitoring 

Dear Dr. Xiao:

I'm pleased to inform you that your manuscript has been deemed suitable for publication in PLOS ONE. Congratulations! Your manuscript is now with our production department. 

Kind regards, 

on behalf of

Professor Pasquale Avino 

Academic Editor

PLOS ONE